# Shining light on the microscopic resonant mechanism responsible for cavity-mediated chemical reactivity

Christian Schäfer [1,2,3,4] ✉, Johannes Flick [5,6,7,8] ✉, Enrico Ronca [9] ✉, Prineha Narang [6,10] ✉ & Angel Rubio [1,2,5] ✉

Strong light–matter interaction in cavity environments is emerging as a promising approach to control chemical reactions in a non-intrusive and efficient manner. The underlying mechanism that distinguishes between steering, accelerating, or decelerating a chemical reaction has, however, remained unclear, hampering progress in this frontier area of research. We leverage quantum-electrodynamical density-functional theory to unveil the microscopic mechanism behind the experimentally observed reduced reaction rate under cavity induced resonant vibrational strong light-matter coupling. We observe multiple resonances and obtain the thus far theoretically elusive but experimentally critical resonant feature for a single strongly coupled molecule undergoing the reaction. While we describe only a single mode and do not explicitly account for collective coupling or intermolecular interactions, the qualitative agreement with experimental measurements suggests that our conclusions can be largely abstracted towards the experimental realization. Specifically, we find that the cavity mode acts as mediator between different vibrational modes. In effect, vibrational energy localized in single bonds that are critical for the reaction is redistributed differently which ultimately inhibits the reaction.

In recent years, strong light-matter interaction[1] has experienced a surge of interest in chemistry and material science as a fundamentally new approach for altering chemical reactivity and physical properties in a non-intrusive way[2–6]. Seminal experimental and theoretical work has illustrated the possibility to control photochemical reactions[7–12] and energy transfer[13–20], strongly couple single molecules[21–24] or extended systems[25–27]. Vibrational strong-coupling is a particularly striking example. For instance, it was

observed that coupling to specific vibrational excitations can inhibit[28–30], steer[31], and even catalyze[32] a chemical process at room temperature. While experimental work continues to make strides, a theoretical understanding of the microscopic mechanism that controls chemical reactions via vibrational strong-coupling still remains largely unexplained. Initial attempts to describe vibrational strong-coupling in terms of equilibrium transition-state theory[33–35] have suggested no resonant dependence on the cavity frequency, in

[1]Max Planck Institute for the Structure and Dynamics of Matter and Center for Free-Electron Laser Science & Department of Physics, Hamburg, Germany. [2]The Hamburg Center for Ultrafast Imaging, Hamburg, Germany. [3]Department of Physics, Chalmers University of Technology, Göteborg, Sweden. [4]Department of Microtechnology and Nanoscience, MC2, Chalmers University of Technology, Göteborg, Sweden. [5]Center for Computational Quantum Physics, Flatiron Institute, New York, NY, USA. [6]John A. Paulson School of Engineering and Applied Sciences, Harvard University, Cambridge, MA, USA. [7]Department of Physics, City College of New York, New York, NY, USA. [8]Department of Physics, The Graduate Center, City University of New York, New York, NY, USA. [9]Istituto per i Processi Chimico Fisici del CNR (IPCF-CNR), Pisa, Italy. [10]Physical Sciences, College of Letters and Science, University of California, Los Angeles, Los Angeles, CA, USA. ✉e-mail: christian.schaefer.physics@gmail.com; jflick@flatironinstitute.org; enrico.ronca@pi.ipcf.cnr.it; prineha@ucla.edu; angel.rubio@mpsd.mpg.de

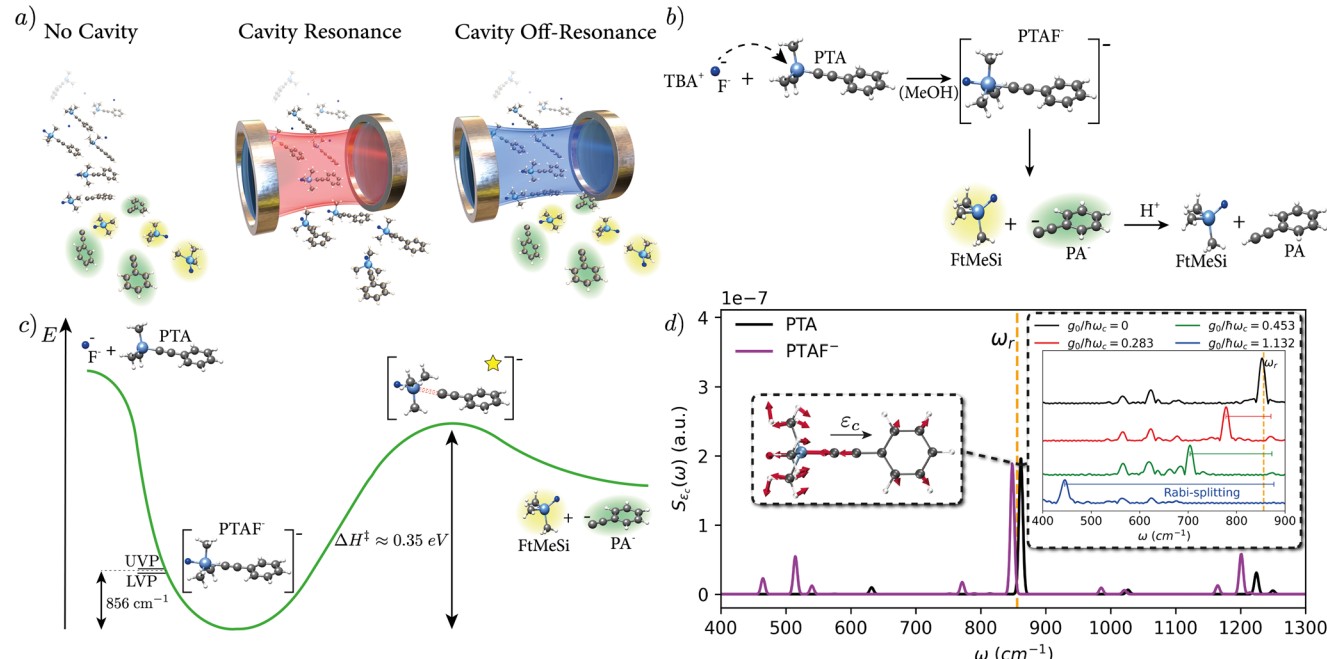

**Fig. 1 | Resonant vibrational strong-coupling can inhibit chemical reactions.** **a** Strong resonant (red shaded) coupling between cavity and vibrational modes can selectively inhibit a chemical reaction, i.e., preventing the appearance of products (shaded green and yellow), that is present under off-resonant (blue shaded) conditions or outside the cavity environment. **b** Illustration of the reaction mechanism for the deprotection of 1-phenyl-2-trimethylsilylacetylene (PTA), with tetra-n-butylammonium fluoride (TBAF) and **c** energetic of the reaction in **b** in free-space (further details see text). The successful reaction involves breaking the Si-C bond and thus overcoming a transition-state barrier of 0.35 eV. The upper (UVP) and lower (LVP) vibrational polariton splitting is comparably small to the transition-state (yellow star) enthalpy $\Delta H^{\ddagger}$. **d** Vibrational absorption spectrum along the cavity polarization direction $S_{\varepsilon_c}(\omega) = 2\omega \sum_{j=1}^{N_{vib}} |\varepsilon_c \cdot \mathbf{R}(\omega_j)|^2 \delta(\omega - \omega_j)$ illustrating the strong-coupling of the vibrational eigenmode at 856 cm$^{-1}$ (yellow-dashed vertical line) with the cavity polarized along $\varepsilon_c$ for PTAF$^-$ (magenta) and the isolated PTA complex (black). The insets show the coupled vibrational mode of PTA and the light-matter hybridization under vibrational strong-coupling. Our time-dependent calculations describe the correlated (non-adiabatic) movement of electrons, nuclei and cavity field during the reaction. The strong asymmetry between lower and upper polariton originates from the high coupling and the related interplay of electronic, nuclear and self-polarization contribution (see Supplementary Fig. 2). The cavity frequency will be changed between 43 and 1584 cm$^{-1}$ in order to investigate resonant effects.

stark contrast with the experimental observations. A second approach suggesting an effective dynamical caging by the cavity[36] partially introduced frequency dependence, but so far has been unable to connect to the experimentally observed frequency dependence.

In this work, we provide the first ab initio study of cavity mediated chemical reactivity that directly addresses and qualitatively recovers the experimental observations[28,29,31]. The fully correlated dynamic between nuclei, electrons and photons allows us to draw important conclusions about the microscopic resonant mechanism responsible for cavity-mediated chemical reactivity. Specifically, we observe the inhibition of the deprotection reaction of 1-phenyl-2-trimethylsilylacetylene (PTA) presented in ref. 28 and schematically illustrated in Fig. 1. Our ab initio investigations obtain the resonant conditions observed in experiment without the presence of a solvent, thus resolving a recent debate questioning previous experimental investigations[37,38]. Importantly, our calculations support the hypothesis that coupling vibrational modes involving the reactive atoms modifies the chemical reactivity. This has been partially indicated by experiments[28,31]. At the same time, we recover critical features of previous theoretical investigations, including the need to go beyond classical transition-state theory and the appearance of dynamical caging effects, which allows to deduce their relevance for experimental realizations. While further investigations will be necessary to entirely resolve the effects behind vibrational strong-coupling, we provide critical theoretical evidence to settle an active debate in the field[28,33–38], suggesting that strong-coupling to specific vibrational modes can indeed modify chemical reactivity. Our approach relies on the recently introduced quantum-electrodynamical density-functional theory (QEDFT) framework[23,39–41] that enables the full

description of electronic, nuclear and photonic degrees of freedom from first principles. QEDFT recovers the resonant dynamic nature of the chemical inhibition under vibrational strong-coupling, illustrating the strength embodied by first-principle approaches in this context. We find that the cavity introduces a new pathway to redistribute vibrational energy during the reaction. Energy deposited in a single bond during the reaction quickly spreads to the set of correlated modes such that the probability to break a specific bond is diminished. Consequentially, if the coupled mode possesses a substantial Si-C stretching character, relevant in the dissociation step, the reactivity is inhibited.

## Results

### Reaction mechanism and resonant vibrational strong-coupling from first-principles

Under typical reaction conditions, that is, no vibrational strong-coupling, the deprotection reaction of 1-phenyl-2-trimethylsilylacetylene (PTA) with tetra-n-butylammonium fluoride (TBAF) to give phenylacetylene (PA) and fluorotrimethylsilane (FtMeSi) is expected to evolve (see Fig. 1b) as follows[28,42]. In solution (usually methanol) the F$^-$ ions released by TBAF form an intermediate pentavalent complex (PTAF$^-$) that reduces considerably the barrier for the dissociation of the Si-C bond, inducing the exit of the phenylacetylide anion (PA$^-$). In solution a fast protonation of the PA$^-$ brings to the formation of the final phenylacetylide (PA) product. Here, we will describe the explicit evolution of the reaction for a single molecule represented by an ensemble of thermally distributed independent trajectories that follow the equations of motion provided by (quantum-electrodynamical) density-functional theory. Our numerical investigations start from the PTA+F$^-$ initial state over the intermediate pentavalent PTAF$^-$ complex up to the breaking of

the Si-C bond (see Fig. 1c), thus explicitly including the rate limiting step (PTA+F⁻ → FtMeSi + PA⁻).

In order to simulate vibrational strong coupling inside the cavity and its effect on the reaction, we couple a single cavity mode with variable frequency $\omega_c$, effective cavity volume $V_c$, and fixed polarization $\boldsymbol{\varepsilon}_c$ to the molecular dipole moment $\hat{\mathbf{R}}$ according to

$$\hat{H} = \hat{H}_{\text{Matter}} + \hbar\omega_c(\hat{a}^\dagger\hat{a} + \tfrac{1}{2}) + \sqrt{\frac{\hbar\omega_c}{2\varepsilon_0 V_c}}(\boldsymbol{\varepsilon}_c \cdot \hat{\mathbf{R}})(\hat{a}^\dagger + \hat{a}) + \frac{1}{2\varepsilon_0 V_c}(\boldsymbol{\varepsilon}_c \cdot \hat{\mathbf{R}})^2 {}^{40,43}.$$

The dimensionless ratio $g_0/\hbar\omega_c$ with coupling $g_0 = ea_0\sqrt{\hbar\omega_c/2\varepsilon_0 V_c}$ provides an indication for relative light-matter coupling strengths. Including experimental cavity losses into the theoretical description results in minor changes as shown in the SI. In order to limit the considerable computational cost of our calculations, we will focus in the following on a subset of quickly reacting trajectories (details in Methods 4). This preferential selection implies that the cavity has to exert sizeable effects on the single molecule within a short time-frame. Since this timescale is correlated with the light-matter coupling strength, we require also a sizeable (enhanced) light-matter coupling strength in our simulations[44]. Increasing the coupling strength further inhibits the chemical reaction (see Supplementary Fig. 7), we observe the same trend as in experiment[28]. The specific value chosen here does not influence the qualitative observation, as discussed in the SI, but dominantly determines the strength of the influence of the cavity on the reaction. We will discuss in the following how such local effective couplings could emerge and their relation to experiment. Retaining the quadratic operator (self-polarization) $\frac{1}{2\varepsilon_0 V_c}(\boldsymbol{\varepsilon}_c \cdot \hat{\mathbf{R}})^2$ is critical to ensure a physically sound result[43]. The correlated evolution of electronic, nuclear, and photonic system is described by quantum-electrodynamic density-functional theory using Ehrenfest's equation of motion for nuclear and Maxwell's equation for photonic degrees of freedom[40] (details in Methods 4).

A prerequisite for reactive control is the strong-coupling condition, that is, a hybridization energy between vibration and cavity larger than the combined decoherence of the system. Intuitively, the cavity excitation would be overlaid with a vibrational excitation and the hybridization correspondingly becomes visible. Obtaining vibro-polaritonic spectra and confirming the strong-coupling condition is thus a first essential step in any ab initio QED chemistry investigation. Fig. 1d illustrates a subset of the vibrational absorption spectrum along the cavity polarization axis for isolated PTA and pentavalent PTAF⁻ complex. A clear vibrational resonances is observed around $\omega_r = 856\,\text{cm}^{-1}$, suggesting that the vibrational strong-coupling conditions are met around the local minimum (PTAF⁻) as well as for all molecules previous to any reaction (PTA). Surely, many other vibrational resonances are available.

During the reaction, however, the molecular geometry and consequently its vibrational spectrum change considerably (more details in the SI) such that simulations involving a single molecule are unlikely to exhibit a sharp resonant condition for an effect of the cavity on the reaction. In addition, the Si-C bond of special interest contributes to a set of vibrational eigenmodes which are distributed over a wide frequency-range. In combination, retaining exact resonance between a specific cavity mode and a specific vibrational mode during the reaction is virtually impossible but also not necessary. Our investigations illustrate that reduced model descriptions, while powerful and intuitive in many situations, can be misleading at times and that first principles calculations are essential in the future understanding and development of polaritonic chemistry.

In order to identify the physical microscopic mechanism behind the cavity induced inhibition of the Si-C bond-breaking, we perform extensive real-time quantum-electrodynamical density-functional theory calculations for a set of 30 trajectories which are launched with initial conditions sampled from a thermal distribution at 300 Kelvin. A subset of 8 trajectories undergoing the reaction outside the

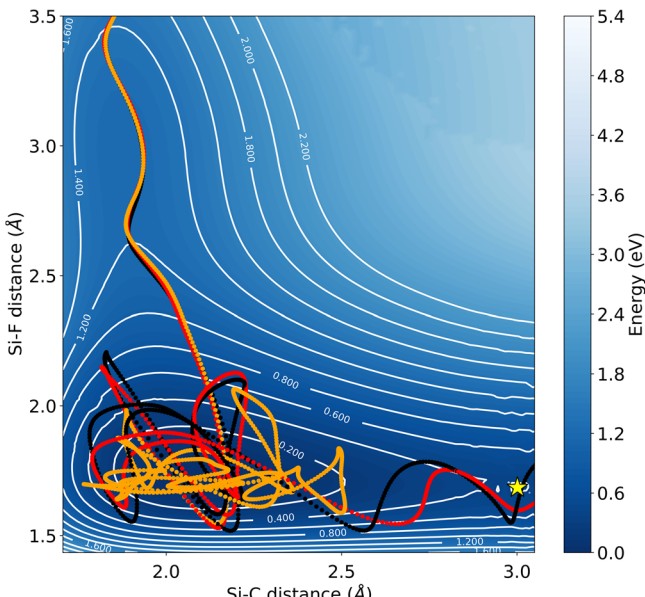

**Fig. 2 | (Non)reactive trajectories under vibrational strong-coupling.** Exemplary trajectory outside $g_0/\hbar\omega_c = 0$ (black), inside the cavity on resonance $g_0/\hbar\omega_c = 1.132$, $\omega_c = 856\,\text{cm}^{-1}$ (orange) and off-resonant $g_0/\hbar\omega_c = 1.132$, $\omega_c = 1712\,\text{cm}^{-1}$ (red) undergoing the reaction illustrated in Fig. 1b, c. The transition-state is indicated with a yellow star. Encoded in the transparency is the relative angle between Si-C and cavity polarization axis (inset Fig. 1d). The molecular axis remains largely oriented along the cavity polarization during the reaction.

cavity will be used for all further analysis. The high computational cost of the current framework limits the number of trajectories that we can address in the statistical ensemble. This demands to sample more densely around highly reactive trajectories which shortens the average reaction-speed. As a consequence, the light-matter coupling strength is artificially increased to enhance cavity-mediated phenomena at those shorter time-scales. Smaller coupling values provide similar predictions (see Supplementary Fig. 7) but would require more computational resources to resolve the cavity influence. The subsequent investigations provide thus strong indications of cavity modified reactivity but are certainly limited in their statistical significance. Fig. 2 illustrates one such exemplary trajectory in free-space (black), inside a off-resonant cavity (red) and when strongly coupled to the resonant cavity-mode (orange). The potential energy surface is shown here for illustrative purposes only and has not been employed in the propagation. For the given set-up, the way F⁻ attaches and how the pentavalent PTAF⁻ complex is formed is barely disturbed. Once the molecule enters the pentavalent complex, a clear change in the dynamics of the Si-C bond is observed in resonance. The resonant cavity effectively traps the system around the local minimum, extending the reaction-time beyond the simulation time of 1 ps by protecting the Si-C bond from dissociating.

To further elucidate the bond-strengthening feature of the cavity, we project the dynamical nuclear coordinates during the time-dependent evolution on the vibrational eigenmodes of the bare PTAF⁻ system (without cavity). Fig. 3 shows the difference of mode occupation between strong resonant coupling $\omega_c = 856\,\text{cm}^{-1}$ and far off-resonant coupling $\omega_c = 43\,\text{cm}^{-1}$. Particularly relevant is the time-domain in which the F anion attaches to PTA at 200 fs and the Si-C bond-breaking in free-space at 500 fs. Without surprise, many (anharmonic) vibrational modes contribute to the reaction, rendering it challenging to describe the reaction in terms of a reduced subset of coordinates. While the cavity has an overall minor effect on the vibrational occupations, it leads to considerable redistribution of occupation in modes along the F-Si-C-C chain. For instance, between

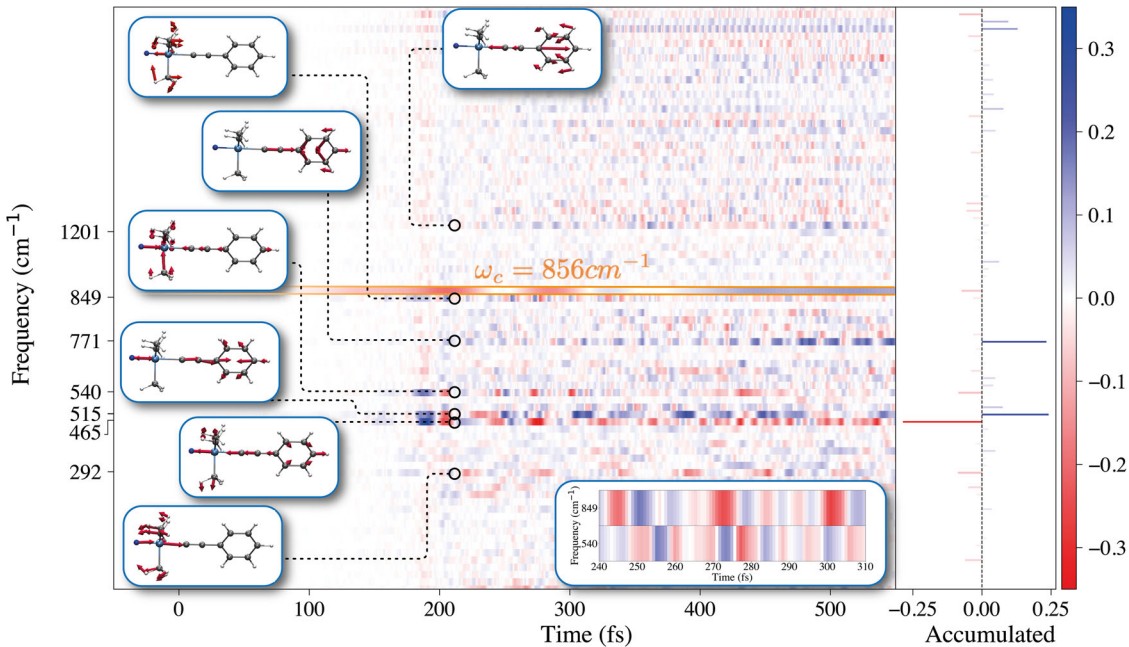

**Fig. 3 | Cavity-mediated redistribution of vibrational energy.** Time-resolved influence on the mode occupations by resonant vibrational strong-coupling. Illustrated is the trajectory averaged difference in normalized mode-occupation between on resonant condition $\omega_c = \omega_r = 856$ cm$^{-1}$, $g_0/\hbar\omega_c = 1.132$ and far off-resonance $\omega_c = 43$ cm$^{-1}$, $g_0/\hbar\omega_c = 1.132$ for all 8 trajectories that undergo the reaction outside the cavity. The bar-plot shows the accumulated difference divided by a factor of 100. The cavity mode (bordered orange) is here represented by the difference in normalized mode-displacement $q(t) = \sqrt{\hbar/2\omega_c}\langle\hat{a}^\dagger + \hat{a}\rangle$ and re-scaled by the factor 1/4. Insets illustrate relevant normal modes or highlight the correlated occupancy between modes 849 and 540 cm$^{-1}$.

240 and 310 fs, the cavity mode induces a coherent exchange of occupation between the coupled mode at 849 cm$^{-1}$ and the vibration at 540 cm$^{-1}$. This coherent exchange seems to originate from the structural similarity between the involved vibrational modes, both featuring strong methyl and Si-F contributions. Si-F and Si-C bond are correlated with a strength that is slightly decreasing under resonant light-matter coupling (see Supplementary Fig. 5). Interestingly, the strongest changes appear for the modes at 771, 515, and 465 cm$^{-1}$, all characterized by strong F-Si-C contributions and with minor or no methyl contribution. Resonant coupling between cavity and the mode at 849 cm$^{-1}$ is thus altering foremost the energy exchange within the F-Si-C-C bond-chain. To summarize, the cavity can prevent the bond-breaking of Si-C by redistributing energy into other vibrations along the F-Si-C-C chain. This effectively correlates the vibrations along the chain protecting the Si-C bond. In a similar spirit, Erwin et al.[45] demonstrated that strong coupling can alter the anharmonic coupling between vibrational excitations while Sun and Vendrell[46] recovered the effect of cavity-mediated energy-redistribution for the cis-trans isomerization of HONO. The dominant effect of vibrational strong coupling on a ground-state chemical reaction would be therefore that it provides means to alter the redistribution of energy from a specific bond into other degrees of freedom. Particularly relevant for this mechanism is the Si-C character of the coupled vibration as will be shown in the following.

Our conclusions may be viewed in the light of altering the vibrational contribution to Polanyi's rule[47,48]. For a reaction A+BC → AB+C, a transition-state close to the product, as present on our PES, implies a strong dependence on the vibrational energy along the reactive path. The cavity minimizes now vibrations in the Si-C bond and thus reduces the effective vibrational energy along the reactive path. As a consequence, the reaction is inhibited and becomes increasingly temperature dependent. While this simplified mechanistic explanation can surely only serve as a rule of thumb, it might suggest a consistent approach to further uncover the role and potential of vibrational strong coupling. Furthermore, these observations suggest a clear dependence on the symmetry of the molecule and associated reaction pathways, in conceptually close agreement with recent experimental investigations[49]. We note that, we decided here to compare between resonant cavity and strongly detuned cavity in order to eliminate the potential influence of self-polarization contributions that are frequency independent and become relevant in the ultra-strong coupling domain (see e.g. Supplementary Fig. 2).

We acknowledge that such correlations via the cavity-mode would necessarily depend on the specific chemical complex, reaction mechanism and ambient conditions. Our results and experimental observations[28,29,31] suggest, however, that tuning the cavity in resonance to a molecular vibrational excitation with relevant contribution to reactive bonds prefers to inhibit typically dominant reaction pathways. In our investigations, the inhibition of the reaction for a given temperature can be overcome by increasing the molecular temperature to the point that the energetic redistribution mediated by the cavity becomes negligible. Each pathway affected by the cavity becomes increasingly sensitive to thermal changes. This provides room for other usually suppressed pathways, leads to an overall tilt in the reactive landscape, and alters the thermodynamic characteristics of the reaction. Whether catalytic effects via vibrational strong-coupling to solvents[32] follow the same rationale and pathway remains an open question.

The experimental conditions in refs. 28, 29, 31, 32 suggest an involvement of a macroscopic number of molecules $N$ that collectively couple to the cavity excitation, effectively representing a set of synchronized oscillators leading to a $\sqrt{N}$ enhancement of the effect. Such a collective interaction clearly shows up in an optical measurement but it does not necessarily imply that a single molecule is strongly affected by the light-matter interaction as the local contribution in a collective state will scale with $1/\sqrt{N}$.

However, the individual molecule undergoing the reaction will continuously change its vibrational structure and no longer behave as the large ensemble of identical collectively coupled reactant molecules. This might call for the introduction of a more local interpretation of

collective strong coupling in chemistry, distinguishing between the reacting molecule and the collective environment in the spirit of an impurity problem[44,50]. Indeed, recent theoretical work suggests that collective strong coupling between a set of identical molecules and a single reacting molecule can result in stronger contribution of the reacting molecule to a polaritonic state. As a consequence, the polaritonic state can obtain a stronger local character than what we would expect from a simplified model with $N$ identical emitters contributing with $1/\sqrt{N}$ to the polaritonic wave function[5,44,50]. Furthermore, Coulomb mediated correlation, which we can expect to be non-negligible in solution, can result in locally enhanced dipole moments that effectively magnify the light-matter interaction strength[51]. It should be noted that collective strong coupling remains an essential prerequisite for those effects to play any role on the experimental scale. In this work we focus on the single-molecular strong coupling and acknowledge that the precise interplay between collective ensemble and local chemistry represents an open problem in the field.

In addition to the here observed intramolecular energy redistribution, intermolecular interactions mediated by the cavity might further inhibit accumulation of energy in specific bonds, thus preventing specific reactions. Recent molecular-dynamics force-field simulations addressing the energetic relaxation of $CO_2$ molecules suggest indeed that the cavity can facilitate intermolecular redistribution of vibrational energy[20]. Experimental ambient conditions influence the coherence of those processes and will likely determine the relevance of intra- and intermolecular vibrational energy redistribution. Further, intra- and intermolecular processes should obey a different dependence on the symmetry of the molecule. The combination of both 'handles' could provide a possible path to elucidate their individual relevance in vibrational strong-coupling.

## Observation of a resonant effect

While experiments so far have indicated a clear dependence on the resonant conditions between vibration of interest and the cavity mode, theory has not been able to agree with this as a critical prerequisite. Hidden in the overall debate are two aspects that are combined in a single experimental observation. The first is the strong-coupling condition itself. When decoherences are stronger than the light-matter interaction no hybridization and subsequently no relevant effect on the chemical reaction can be observed[28]. Second, the vibrations that dominantly contribute to the bond-breaking have to be involved in this strong-coupling condition.

In our ab initio theoretical investigation, the strong-coupling condition is always fulfilled, and thus we dominantly investigate the second resonant condition. By varying the length of the idealized Fabry-Pèrot cavity, we tune the frequency of the cavity mode while keeping the ratio between coupling and frequency $g_0/\hbar\omega_c$ constant. Fig. 4 illustrates our extensive ab initio calculations addressing the resonant effect in a compact trajectory averaged form. We show the trajectory averaged Si-C distance over time (a), the time-averaged Si-C distance (b), the spectrum along the cavity polarization of PTAF$^-$ and the spectrum weighted with its Si-C contribution (c).

All trajectories that undergo the reaction outside the cavity (dotted black lines) show a diminished reactivity inside the cavity. Clear resonant features appear around 86 (red dashed-dotted vertical line), 290, 570, and 1300 cm$^{-1}$ in addition to a broad shoulder between 600 and 1000 cm$^{-1}$ that includes the experimentally investigated resonance at 856 cm$^{-1}$ (orange dashed vertical line). Setting Fig. 4b in relation to the vibrational spectrum in Fig. 4c illustrates two different mechanisms. At 86 cm$^{-1}$ (red dashed-dotted vertical line), no vibrational spectral density exists but the energy coincides with the curvature of the potential energy surface at the transition-state (see Supplementary Discussion 3). This leads to a dynamical caging effect of the trajectory around the transition-state, in agreement with recent findings by Li et al.[36] and discussed in more detail in the SI, that

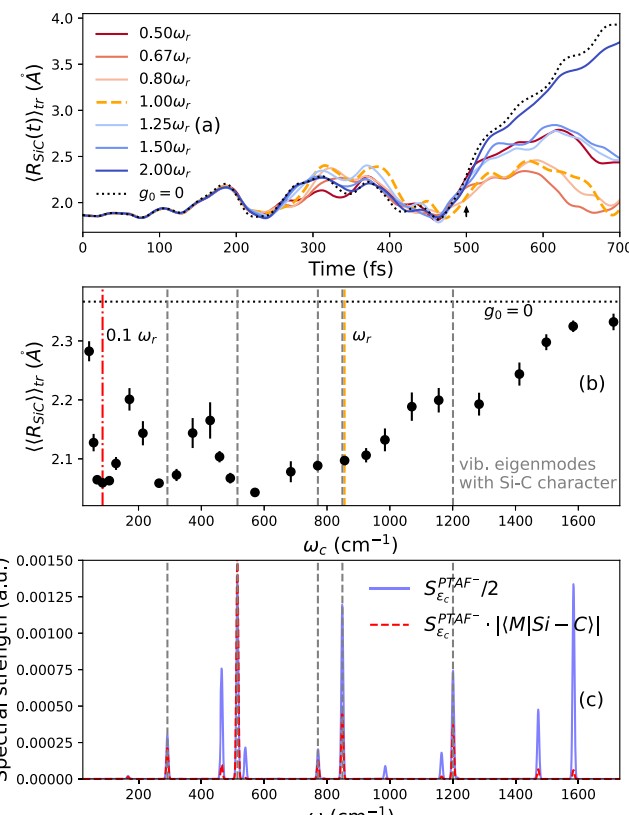

**Fig. 4 | Frequency dependency of cavity-mediated chemical inhibition.** Time-resolved trajectory averaged Si-C distance **a**, time-averaged Si-C distance $\langle\langle R_{SiC}\rangle\rangle_{tr}$ **b**, and (Si-C weighted) spectral strength in polarization direction **c** for varying cavity frequency $\omega_c$ relative to the resonant frequency $\omega_c = \omega_r = 856$ cm$^{-1}$. The dashed vertical lines indicate the position of the vibrational eigenmodes with sizeable Si-C character and spectral strength $S_{\epsilon_c}^{PTAF^-}$. The ratio $g_0/\hbar\omega_c = 1.132$ is kept constant. Error-bars show the standard error of the trajectory average. We use all 8 trajectories that show the reaction outside the cavity. In free-space (black dotted), the reactive trajectories will start to break the Si-C bond at around 500 fs (indicated by the black arrow). The red dashed-dotted vertical line emphasizes the resonance predicted by Grote-Hynes theory (see text for more details). The cavity has the overall tendency to prevent the formerly reactive trajectories from breaking the Si-C bond. This effect shows a complex frequency dependence that includes two separate regimes as explained in the text. The Si-C-stretching weight $|\langle M|Si-C\rangle|$ was obtained by projecting the normalized Si-C stretching on the vibrational modes.

prevents the chemical reaction. No experiment investigated this frequency domain so far such that this mechanism seems unrelated to the experimental realizations. The lack of vibrational spectral density in this domain would prevent the occurrence of polaritonic states. The second mechanism involves all vibrational modes with a sizeable Si-C-stretching contribution (c, red dashed line), indicated by grey-dashed vertical lines in (b) and (c). Comparison with Fig. 3 clarifies that the highlighted vibrational modes play a central role in the reaction mechanism and are strongly affected by the cavity. Intuitively, stronger Si-C character in the polaritonic states will affect the Si-C bond stronger, resulting in the proposed mechanism of a cavity induced vibrational energy redistribution that effectively strengthens the Si-C bond and prevents the reaction. Maximising the light-matter coupling strength via large spectral weight and ensuring a high Si-C character suggests the vibration at 515 cm$^{-1}$ as most promising. The corresponding mode projection is illustrated in the SI Fig. S5. It should be acknowledged however that we find especially the spectral intensity to be sensitive to approximations in the theoretical description (see Supplementary Fig. 4). Larger frequencies imply larger fundamental couplings which in turn result in a stronger contribution of the self-

polarization reflecting in an increased blue-shift (see Supplementary Fig. 2).

The experimental counterpart focused on a small frequency domain around the 856 cm$^{-1}$ vibration[28]. A subsequent publication investigating the influence of collective strong coupling on silyl cleavage reactions for a related structure[31] identified however a clear effect of the cavity when tuned to one out of three vibrations located at 842, 1110, and 1250 cm$^{-1}$. This supports our hypothesis and suggests that vibrational strong coupling could represent an even more flexible tool for chemistry than currently hoped. Let us point out that the single mode approximation to the Fabry-Pérot cavity eliminates the possibility of in-plane momentum for the photon, i.e., the confined electromagnetic field propagates perpendicular to the surfaces of the mirrors. A realistic Fabry-Pérot cavity will feature a quadratic dispersion with respect to the incident angle. In this light, it might surprise that the experiments report any resonant effect[28,30,31]. However, it should be noted, that the polaritonic states quickly approach the limit of bulk polaritons, which exist also in the absence of mirrors, with increasing in-plane momentum[52]. As pointed out by Vurgraftman et al.[52], the concept of cavity mediated reactivity considers the change when enclosing the system inside a cavity which implies the strongest effect for zero in-plane momentum. While the magnitude of the effect described by Vurgraftman et al. is small, it unambiguously demonstrates the special role of the lowest harmonic eigenmode with zero in-plane momentum. A noticeable difference between our theoretical calculations and experiments is the broadness of the observed resonant features. As in the present work, we are computationally limited to a single molecule and a small set of trajectories. Energy redistribution can appear then only between modes in the same molecule (intramolecular energy redistribution). When larger ensembles of molecules collectively couple to the cavity intermolecular energy redistribution will likely provide additional and potentially more efficient channels for energy redistribution. In order to observe a significant effect for a small set of trajectories it was necessary to select quickly reacting initial configurations and large light-matter couplings. The corresponding short reaction times and large couplings lead to an influence of the cavity on much shorter time-frames than in experiments and suggest therefore broader frequency features than in experiment. Nevertheless, the qualitative agreement between the here observed mechanism and the experimentally observed features suggests that the here proposed mechanism, i.e., vibrational strong-coupling mediated energy redistribution and subsequent protection of the Si-C bond, largely explains the experimentally observed chemical effect.

## Discussion

Strong light-matter coupling in cavity environments provides a conceptually new and potentially invaluable addition to the current chemical toolbox[2–4]. By leveraging the QEDFT framework, we provide key theoretical insight from first-principles into the inhibition of ground-state reactions under vibrational strong-coupling. Despite disregarding the quantum nature of photons and considering only a single molecule, our results are in good qualitative agreement with experimental observations[28,29,31], suggesting that cavity-mediated chemical reactivity can be largely explained by strong classical light-matter interactions. To the best of our knowledge, we present for the first time the 'in experiment' observed resonant condition.

During the reaction, energy is accumulated in specific bonds which in free-space lead to their dissociation. Under strong vibrational coupling, the cavity-correlated vibrations redistribute energy differently than in free-space, effectively strengthening the here relevant Si-C bond and ultimately inhibiting the reaction. The cavity serves as mediator to other vibrations, in combination, they represent a strongly frequency dependent bath. Our observations are consistent with the recent experimental work by Chen et al.[53] which demonstrated the importance of cavity mediated intramolecular vibrational-energy redistribution for chemical reactivity. Increasing light-matter coupling strength enhances the inhibiting effect of the cavity, we observe a similar trend as in experiment[28]. The resonant condition associated with this microscopic mechanism is apparent and suggests that future theoretical models should focus on an appropriate description of such a frequency dependent bath. Furthermore, our results suggest a sizeable cavity-mediated influence on chemical reactions can be expected when the coupled vibrational mode provides a large oscillator strength and at the same time contributes noticeably to the reaction. Within the here investigated chemical reaction, the cavity provides a way to selectively cool specific vibrations. Recent experiments in which the solvent is strongly coupled[32] suggested that this bath could similarly serve as reservoir, meaning hot solvent molecules transfer energy to the reactant, catalysing the reaction. Our observations are in line with the experimentally observed relevance of the symmetry of the coupled vibrational mode[49] as the energy redistribution between vibrations due to the cavity demands a consistent alignment between vibration and cavity polarization, suggesting that different symmetries will give rise to different reshuffling of energy. Even in absence of collective effects we recover the sought resonant conditions[28,29,31]. The limitations of the current approach, namely, the single molecule description with enhanced light-matter coupling strength, result in considerably broader resonances than experimentally observed. A connection to thermodynamic quantities could not be established. While intermolecular interactions could contribute to the energy redistribution responsible for the cavity mediated effect on chemical reactivity, the agreement between our theoretical predictions and experimental observations suggests that the mechanism will remain qualitatively similar. It seems however likely that intermolecular energy redistribution could enhance the influence of the cavity. Our observation serves as primer for the understanding of the local microscopic changes in chemical reactions induced by strong light-matter coupling in cavities. Future investigations will explore in more detail how collective effects can modify our conclusions, a critical aspect that remains a matter of active debate[5,28,44,46,50,54–57].

In agreement with previous studies[33–36,46,55,58,59], we conclude that dynamic features play a much more pronounced role under strong light-matter coupling than commonly assumed by standard transition-state-theory for ground-state chemical reactivity. Our ab initio description of matter is parameter-free and thus suggests that our conclusions can be extrapolated to similar reactions, providing a precursor to a new field of theoretical ab initio polaritonic chemistry. Experiments with focus on a consistent investigation and control of the transition-state position, i.e., the realms of Polanyi's rule[47], would provide valuable information about the underlying mechanism. Handles such as symmetry and coherence could elucidate to which extend inter- and intramolecular energy redistribution contribute to vibrational strong-coupling. How such kinetically driven effects can persist into the realm of realistic ambient conditions for large ensembles of reacting molecules remains a key question necessitating further theoretical and experimental investigations where the present findings provide the seed to guide them.

## Methods

The potential energy surface is obtained using the ORCA code[60] with the 6-31G* basis set, employing DFT and the PBE functional. The time-dependent calculations were performed using the real-space TDDFT code Octopus[61]. Building the pentavalent complex is entropically unfavored and unlikely such that just a very limited set of trajectories would result in a successful attachment of the F anion. To reduce the amount of non-reactive trajectories we kicked the complex out of its local minimum in the pentavalent configuration along the minimal energy path running the reaction backwards towards the initial condition to estimate the collision speed and angle leading to the reaction. We obtained ideal angles between Si-C and Si-F bond of

around 180 degrees. An alternative angle for $F^-$ to attack Si is at 60 degrees but the latter demands higher temperatures on the order of 900K. Using the linear configuration, we obtained first initial geometries from Orca with a subsequent further optimization in the Octopus code using a constrained force minimization for a fixed Si-F distance of 4 Å. According to Fig. 1 of the main text, the initial $F^-$+PTA configuration is energetically higher than the transition-state barrier of 0.35 eV such that a few trajectories will undergo the reaction within a reasonable time-frame while the majority will remain trapped in the local minimum of the PTAF$^-$ configuration. Starting from those initial geometries, we sample initial velocities according to a Bolzmann distribution at 300 K, i.e., a Gaussian normal distribution with nuclei specific variance $\sigma_{v_i} = \sqrt{k_b T / M_i}$. Furthermore, we point the F momentum towards the Si and remove the center of mass movement. We propagate 10 such trajectories and obtain a single reactive trajectory at 300K shown in fig. 2. Then, in order to limit the necessary number of trajectories while still obtaining a reasonable number of reactive trajectories, we sample around the first reactive trajectory a set of 20 additional trajectories with a relative temperature of 20 K, removing again the center-of-mass momentum. We find in total 8 reactive trajectories outside the cavity. If not further specified the illustrated observables represent the average of those 8 trajectories. Inside the cavity, the initial photon-mode displacement is provided by the zero-electric field condition $q(t_0) = -\frac{\lambda}{\omega} \cdot \mathbf{R} \leftrightarrow \mathbf{E}(t_0) = 0$ with $\dot{q}(t_0) = 0$ and $\mathbf{R} = \sum_i^{N_n} Z_i \mathbf{R}_i - \sum_i^{N_e} \mathbf{r}_i$ [40,43,62]. The polarization of the cavity was chosen along the Si-C axis which coincides with the x-axis. This initial configuration is then propagated in the Octopus code[61] using the enforced time-reversal symmetry (electronic subspace) + velocity verlet (nuclear subspace) time-stepping routine with electronic $\Delta t_e = 0.0012/eV$, photonic $\Delta t_p = 10\Delta t_e$ and nuclear $\Delta t_n = 10\Delta t_e$ timesteps. Photonic and nuclear coordinates are accelerated by a factor $10$ [40,63] in order to shorten the calculation time. For the electronic subspace, we use the revPBE[64] electronic exchange and correlation potential and the standard SG15 Pseudo-potential set. The nuclear coordinates are moving according to the classical Ehrenfest equations of motion, i.e., electronic and nuclear system are self-consistently exerting forces on each other. The photonic coordinates are coupled via the classical Ehrenfest light-matter coupling using the QEDFT framework which leads to classical cavity fields acting on nuclear and electronic degrees of freedom[23,39–41]. More details on the computational approach can be found in the SI. We use a numerical box of $V = (24a_0)^3$ and a grid spacing of $0.24a_0$ with the bohr radius $a_0$. The parameters of the grid are chosen such that numerically caused changes in the energy are minor compared to thermal energies. We monitor the time-evolution of the complex and define the Si-C bond as broken when the Si-C distance increases beyond the transition-state. The spectra showing the Rabi-splitting in Fig. 1d have been obtained by propagating for 3 ps on a grid that combines spheres of size 6 Å around each atom with a grid spacing of $0.24$ $a_0$. Here, the nuclei are initialized with random velocities corresponding to 50 K.

## Data availability

The data as well as input and plotting scripts that support the findings of this study are available on the public repository https://gitlab.com/jflick/octopus-fork-public or from the authors upon request.

## Code availability

The numerical QEDFT implementation is part of the Octopus code of which a public fork is available https://gitlab.com/jflick/octopus-fork-public. Octopus code and ORCA are open source.

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

## Acknowledgements

We thank Anoop Thomas, Michael Ruggenthaler, and Göran Johansson for insightful discussions. The Flatiron Institute is a division of the Simons Foundation. This work was supported by the European Research Council (ERC-2015-AdG694097) [AR], the Cluster of Excellence 'CUI: Advanced Imaging of Matter' of the Deutsche Forschungsgemeinschaft (DFG)—EXC 2056—project ID 390715994 [AR], Grupos Consolidados (IT1249-19) [AR], partially by the Federal Ministry of Education and Research Grant RouTe-13N14839 [AR], the SFB925 "Light induced dynamics and control of correlated quantum systems" [AR], the Swedish Research Council (VR) through Grant No. 2016-06059 [CS], the Department of Energy,

Photonics at Thermodynamic Limits Energy Frontier Research Center, under Grant No. DE-SC0019140 [PN]. P.N. gratefully acknowledges a Moore Inventor Fellowship through Grant GBMF8048 from the Gordon and Betty Moore Foundation and support from the Canadian Institute for Advanced Research (CIFAR) BSE Program.

## Author contributions

C.S., J.F., and E.R. contributed equally. A.R. and C.S. conceived the project. C.S., J.F., and E.R. obtained, evaluated and interpreted the data. C.S. prepared a first draft, C.S. and E.R. prepared the figures with input from all authors. All authors discussed the results and edited the manuscript.

## Funding

## Competing interests

The authors declare no competing interests.
