## [Peer Review File · Nature Communications]

REVIEWER COMMENTS

Reviewer #1 (Remarks to the Author):

This is a resubmission of a manuscript that has been rejected earlier and has now been resubmitted with minor changes.

As far as I can see, the results have not been changed or been extended significantly.

While I think the paper is interesting, I do not think its results do not support the claim of a "comprehensive picture of the microscopic resonant mechanism responsible for cavity-mediated chemical reactivity"

My main points of criticism still hold:

* The observed reaction happens under thermal conditions in solution and the cavity mode is described classically. Under these conditions, the stationary points of the reaction namely, the transition state/barrier height of the combined molecule-cavity system, should give allow an evaluation of a reaction rate. However, this seems not be discussed in the paper.

* The paper mentions a set of 30 trajectories. This seems to be a quite small number, given the size of the system. A statistical analysis of the trajectories would be required here to demonstrate that the conclusions drawn from the trajectories are within the error bars.

* The ab initio trajectory calculations only mention the PCM model to account for the solvent. However, the PCM model is a self-consistent field model and thus seems to be not very suitable for dynamics simulations. It seems more appropriate to include a shell of solvent molecules in the dynamics simulations.

* The manuscript refers to the investigated coupling strength as "strong coupling". However, a coupling strength of $g/h\nu \sim 1$ is rather in the ultra strong regime. It is not clear how the referenced experiment would reach this coupling strength.

* An important point, which is neglected here, are dissipative processes due to imperfect cavity mirrors. It has been shown by several authors in the last two years, that this seems to be of mayor importance for the whole mechanism. This is not discussed in the manuscript. Can the authors rule out that it is not cavity decay that is responsible for removing energy from the relevant bond?

Reviewer #2 (Remarks to the Author):

In this paper, Schafer and co-authors use ab-initio QED simulation to investigate the vibrational strong coupling induced modifications of ground state chemical reactions. Through a direct quantum-electrodynamical density-functional simulation between a single molecule and a single mode of an optical cavity, they observed "multiple resonances behavior", which is, by matching the cavity frequency to the vibrational normal mode frequencies, the Si-C bond distance is reduced, thus, "prohibits" reaction from happening. Based on the simulations, the authors hypothesize that the cavity mode acts as mediator between different vibrational modes, which is the mechanism for the observed bond

pretation mechanism. They further conclude that the vibrational energy localized in single bonds are critical for the reaction to be redistributed differently which ultimately inhibits the reaction.

Despite its limitations (single molecule, no collective effect, single cavity mode, not enough statistics, and so on), to the best of my knowledge, this is indeed the first work that treats the Ebbesen molecule through ab-initio calculations and using dynamics to investigate reactivities. In that sense, I do believe that the VSC community will benefit from such a work. I thus recommend this paper to be published on Nature communication, after the authors addressing the following comments.

(1) Connections with the previous work on vibrational energy redistribution. The mechanism proposed in this work, which is cavity mode acts as mediator between different vibrational modes, might be equivalent to the recent work in [J. Chem. Phys. 156, 014101 (2022)]. In that work, the cavity mode is coupled to a collection of vibrations, which are in turn coupled to the reaction coordinate, thus the cavity mode is effectively mediating energy transfer and influencing reaction rate constants. In that work, through both analytic theory and numerical simulations, it seems that when cavity frequency is tuned to match a frequency that depends on both vibrational frequency and the top of barrier frequency, the reaction rate constant is reduced. I believe that the mechanism is essentially the same as what presented here, which is the cavity mode mediating energy transfer among vibrational modes. In a related work [Angew. Chem. 2021, 133, 15661 -15668], this vibrational energy transfer mechanism is also explicitly explored and confirmed, with a much sharper resonance condition. A detailed comment on the connections between the current work and these two previous works will be extremely helpful.

(2) The limited numerical statistics. The main limitation of the current work is the number of trajectories, which is only 30, and only 8 of them are reactive and were used to analyze the results (Fig. 4b). Even though the time averaging gives a small enough error bar presented in Fig. 4b, the question remaining is how statistically meaningful for those 8 reactive trajectories, especially considering the very limited and specific phase space covered through their initial conditions. Can we trust the results presented in Fig. 4? By adding more trajectories, will all of these resonant features be washed away?

(3) The VSC experiments reported rate constant suppression. The current paper reported a different quantity, which is the bond distance (Fig. 4). To what extent, these two quantities related? I understand intuitively, that makes sense, and the dip in Fig 4 that corresponds to the barrier frequency nicely corresponds to the rate constant reduction of the caging effect reported in Ref. 36. However, can the authors provide a more rigorous argument on the connections between them, which means a shortening of the equilibrium bond distance is guaranteed for a rate constant reduction.

(4) The theoretical approach that authors used is Ref. 40. Based on what I read, it seems to be an Ehrenfest dynamics that treats both nuclear and photonic DOF classically, and only treat the electronic DOF quantum mechanically. If so, this is then different than what is claimed in the main text "Our approach relies on the recently introduced quantum-electrodynamical density-functional theory (QEDFT) framework [23, 39–41] that enables the full description of electronic, nuclear and photonic degrees of freedom from first principles", because it sounds like the author treats everything at the DFT level of the theory, and the classical treatment is usually not considered as "first principle".

(5) What is the difference between this approach in Ref. 40 with the "adiabatic dynamics", which propagates the classical nuclear and photonic trajectories on the adiabatic electronic ground surface, such as used in those earlier studies in Ref.36, in [J. Chem. Phys. 156, 014101 (2022)] and in [Angew. Chem. 2021, 133, 15661 – 15668]. Further, the authors

need to provide more details of this method in either the method section or SI to make the current paper self-contained.

(6) Besides the single molecule limitation, the other major limitation of the current work is that it does not consider cavity loss. Cavity loss explicitly presents in the VSC experiments with microcavities, and it has been theoretically shown to further facilitate the "energy transfer mechanism" between vibrational modes and the cavity mode, in [J. Chem. Phys. 156, 014101 (2022)]. A comment on this will be very helpful.

(7) I would recommend the authors to carefully summarize what agrees with the experimental results and what disagrees with them. For example, the theoretical prediction has a much broader resonance compared to the sharper resonance observed in the experiments, and the experimental frequency at 856 cm⁻¹, which also has a very strong resonance suppression of the rate is missing in the current theoretical prediction.

Reviewer #3 (Remarks to the Author):

The article by Schafer et al. reports QEDDFT/Ehrenfest simulations of an organic silicon deprotection reaction under the influence of a boson mode representing an optical microcavity. This reaction was first studied experimentally in a Fabry-Perot cavity under conditions of collective strong coupling by Thomas et al. The simulation results are valuable contributions to the field. However, as explained in detail below, some of the main claims made in the article are highly questionable when viewed in light of recent works in this field.

1. Previous articles have already shown that the cavity can act to suppress a reaction when the light-matter coupling is many orders of magnitude greater compared to experiments in Fabry-Perot cavities [see e.g., <https://journals.aps.org/prx/abstract/10.1103/PhysRevX.9.021057>, <https://pubs.acs.org/doi/pdf/10.1021/acs.jpcclett.2c00974>, <https://www.sciencedirect.com/science/article/abs/pii/S0301010419311140>, <https://aip.scitation.org/doi/10.1063/5.0006472>]. The challenge, as laid out in e.g., the PRX of Feist et al (as well as in several other works) is to explain why the cavity effect persists under conditions of collective strong coupling, where a large number of molecules couple to the field, the polaritons constitute a tiny fraction of the total number of states, and the single-molecule light-matter coupling is extremely weak (orders of magnitude weaker than those used by the authors).

In a long complicated paragraph on page 4, the authors present a brief discussion of the above point, but it remains very unclear to this reviewer why the effect they reported is expected to persist when a realistic value of the single-molecule light-matter coupling is several orders of magnitude smaller than the valued used in the simulations.

2. In the experiments where reactivity was changed by the cavity, 'off-resonance' only refers to the cavity mode with zero in-plane wave-vector magnitude ($q = 0$), but in general this means that some other cavity mode with q different from zero is on-resonance with the targeted molecular vibration.

Given that these are thermal reactions, there is no obvious reason why a cavity resonance at $q = 0$ with some molecular vibration would be more efficient at suppressing a chemical reaction than a cavity mode with q different from 0.

In the authors' simulations, as in other recent papers, a "resonant effect" is observed, but that could be a mere byproduct of the very extreme assumption that all cavity effects can be

described with a single-photon mode at $q = 0$.

Whether such resonance effect would persist under a more realistic treatment of the optical cavity that accounted for cavity-matter resonance at $q = 0$ is a question that the authors have completely ignored. In fact, there is no discussion of their central assumption that the cavity can be described with a single photon mode when the existence of a continuous spectrum of q different from zero modes in a planar Fabry-Perot cavity is well known. Furthermore, polaritons at incidence angles different from zero are routinely observed and play a key role in photoluminescence, Bose-Einstein condensation, thermal emission, etc.

Given the above arguments, while I agree with the authors that the 'resonant effect' is 'theoretically elusive but experimentally critical', I respectfully disagree that their simulations have explained the mechanism underlying this effect.

In summary, while this is an interesting manuscript, I don't think it provides resolution to the two main challenges of polariton chemistry described above. Therefore, I suggest publication in a more specialized journal (after reformulation of some of the conclusions).

MAX PLANCK INSTITUTE FOR THE STRUCTURE
AND DYNAMICS OF MATTER
Theory Department

MPSD, Luruper Chaussee 149, D-22761 Hamburg

Dr. Margherita Citroni
Nature Communications

Christian Schäfer
Luruper Chaussee 149
D-22761 Hamburg
Phone: +49 40 8998 88332
Fax: +49 40 8998 6570
christian.schaefer.physics@gmail.com

Hamburg, September 7, 2022

Dear Dr. Margherita Citroni,

Attached please find the revised manuscript entitled "Shining Light on the Microscopic Resonant Mechanism Responsible for Cavity-Mediated Chemical Reactivity" by Christian Schäfer, Johannes Flick, Enrico Ronca, Prineha Narang, and Angel Rubio. A detailed response to the reviews (repeated in *italic*) is given below with changes highlighted in red. Additionally, we also attach a version of the manuscript with all highlighted changes.

Reviewer 1:

This is a resubmission of a manuscript that has been rejected earlier and has now been re-submitted with minor changes. As far as I can see, the results have not been changed or been extended significantly. While I think the paper is interesting, I do not think its results do not support the claim of a "comprehensive picture of the microscopic resonant mechanism responsible for cavity-mediated chemical reactivity"

We thank the reviewer for their effort and we appreciate their finding of our paper as 'interesting'. We however respectfully disagree with the reviewer's further assessment. The word 'comprehensive' is certainly debatable and **we decided to reformulate the sentence in order to better represent our work**. While our work is still the only available investigation from first principles that addresses the experiments and finds for the first time clear connections, it is not our intention to claim that this manuscript provides conclusive explanations to all experimental observations. We consider our work a significant step but some question certainly remain to be answered.

*My main points of criticism still hold: * The observed reaction happens under thermal conditions in solution and the cavity mode is described classically. Under these conditions, the stationary points of the reaction namely, the transition state/barrier height of the combined molecule-cavity system, should give allow an evaluation of a reaction rate. However, this seems not be discussed in the paper.*

We thank the referee for raising this important point, which we already discuss thoroughly in the paper. In particular the first paragraph already states: "Initial attempts to describe vibrational strong-coupling in terms of equilibrium transition-state theory [33-35] have suggested no dependence on the cavity frequency, in stark contrast with the experimental observations."

This aspect, i.e., that 'standard' transition-state theory has fundamental limitations in explaining the experimental observations, is then reinforced in many places in the manuscript. For instance, the critical resonant features characterizing the investigated experiment are completely absent if transition-state theory is applied (see e.g. Ref. J. Chem. Phys. 152, 234107 (2020); <https://doi.org/10.1063/5.0006472>). Thus, if a theory is sought that is able to describe experimental observations, it needs to go beyond 'standard' transition state theory, which is one of the main motivations of our work. We would have been happy to consider the reviewer's comment in more detail if a specific reference proving the usefulness of transition-state theory in this context would have been mentioned.

** The paper mentions a set of 30 trajectories. This seems to be a quite small number, given the size of the system. A statistical analysis of the trajectories would be required here to demonstrate that the conclusions drawn from the trajectories are within the error bars.*

We accept that the small set of trajectories is a relevant limitation of our work, a limitation that is clearly stated and discussed in sentences such as "The high computational cost of the current framework limits the number of trajectories that we can address in the statistical ensemble. This demands to sample more densely around highly reactive trajectories which shortens the average reaction-speed.". However, a statistical analysis is illustrated in Fig. 4 (b) and the error-bars are far smaller than all observed effects. We would like to clarify that the more dense sampling around a reactive trajectory improves the statistics considerable – the reaction by itself is very unlikely. This way, we are able to draw more reliable conclusions with a small ensemble, our statistic is therefore weighted towards 'chemically relevant' events, in spirit somewhat related to the Bayesian approach in statistics. This can be nicely seen from the Maxwell-Boltzmann distribution shown below. The histogram resembles the Maxwell-Boltzmann distribution at 300K and counts how many atoms possess a velocity that is within a given range.

Reactive trajectories are highlighted in red. A pronounced high-energy tail is necessary to undergo reaction, our dense sampling increases the probability for such a trajectory to be used within the calculation. It should be noted that even non-reactive trajectories can exhibit strong

Si-C bond-stretchings which contributes additional information to our analysis in Fig. 4. In conclusion, while a much larger ensemble would be certainly desirable, our conclusions are robust within the limited statistical ensemble and, in our opinion, support the claims of the paper.

We added discuss this aspect now in more detail in the SI and provide the here shown Maxwell-Boltzmann distributions.

** The ab initio trajectory calculations only mention the PCM model to account for the solvent. However, the PCM model is a self-consistent field model and thus seems to be not very suitable for dynamics simulations. It seems more appropriate to include a shell of solvent molecules in the dynamics simulations.*

The time-dependent calculations shown in the main manuscript are performed in vacuum. The PCM model has only been used to investigate the influence of solvation on the potential energy surface and the vibrational spectra. As discussed in the main manuscript and especially in the SI, the solvent tends to shift spectral weight but it barely influences the position of vibrational excitations. See e.g. the sentence "It should be acknowledged however that we find especially the spectral intensity to be sensitive to approximations in the theoretical description (see SI)". Our TD-QEDFT calculations without solvent can then be expected to provide adequate results. The relative strength of vibrational peaks and therefore the relative strength of local minima in Fig. 4 (b) could change when the solvent is considered but those aspects are transparently discussed and do not change the overall conclusion.

** The manuscript refers to the investigated coupling strength as "strong coupling". However, a coupling strength of $g/\hbar\omega \approx 1$ is rather in the ultra strong regime. It is not clear how the referenced experiment would reach this coupling strength.*

SI Figure 7 shows that the observed effect scales with the coupling strength. The trend is actually quite consistent with Fig. 3 a in [Angewandte Chemie International Edition 55, 11462 (2016)]. It is therefore clear, that while we use overall larger coupling strengths compared to the experiment, the effects can be expected to persist also at smaller couplings. We discuss the strength of the light-matter interaction and its relation multiple times throughout the manuscript, e.g., on page 4 or in sentences such as "This preferential selection implies that the cavity has to exert sizeable effects on the single molecule within a short time-frame. Since this timescale is correlated with the light-matter coupling strength, we require also a sizeable (enhanced) light-matter coupling strength in our simulations. Increasing the coupling strength further inhibits the chemical reaction (see SI), we observe the same trend as in experiment [28]. The specific value chosen here does not influence the qualitative observation, as discussed in the SI, but dominantly determines the strength of the influence of the cavity on the reaction.". Smaller coupling strength and thus smaller effects would surely demand a much higher statistical resolution. Overall, we thoroughly discuss the consequences of the high coupling strength and illustrate that the conclusions can be expected to remain qualitatively true for lower values.

** An important point, which is neglected here, are dissipative processes due to imperfect cavity mirrors. It has been shown by several authors in the last two years, that this seems to be of mayor importance for the whole mechanism. This is not discussed in the manuscript. Can the authors rule out that it is not cavity decay that is responsible for removing energy from the relevant bond?*

To the best of our knowledge, loss has been identified as relevant for photo-chemical reaction and especially for plasmonic strong coupling where a real external excitation is send into the set-up. Plasmonic systems feature extremely short lifetimes, often on the order of a few fs, due to the quick Landau-damping and hot-carrier generation. This is conceptually very different to the here discussed vibrational strong-coupling in the dark (no illumination) for a ground-state chemical reaction. Furthermore, the experimentally used infrared-cavities feature much higher quality-factors than plasmonic systems and can be considered as coherent on time-scales less then a ps (as investigated in this work). To be specific, Thomas et al. [28] found a FWHM of the cavity eigenmode of 30 cm^{-1} which corresponds via $\tau_{cav} = 1/\Delta f = 1/(\tilde{\nu}c) = 1/(15 \text{ cm}^{-1} \cdot 2.998 \cdot 10^{10} \text{ cm/s})$ to a lifetime of 2.22 ps, which is about 4 times the reaction-time. Cavity-loss is thus much less likely to play a significant role. Nevertheless, in order to support our intuition, **we added losses to the cavity mode in our simulations, mention its effect in the main text, discuss its influence in the SI, and illustrate its relevance below.**

We introduce a friction of the form $2\gamma\dot{q}_\alpha(t)$ to the EOM of the photonic mode and recalculate all reactive trajectories for $\omega_c = 571 \text{ cm}^{-1}$. The used friction values are multiples $\gamma = \{0, 1, 2, 4, 8\} \cdot \gamma_{exp}$ of the experimentally observed value $2\gamma_{exp} = 30 \text{ cm}^{-1}$ or 0.000137 in atomic units.

The influence of photonic friction on the average Si-C bond length depends on time. For short times, we see an increase of the inhibiting effect of the cavity. This short-time effect tends to decrease over time and leads to only minute changes in the time-averaged Si-C distance. Our approximation of a loss-less cavity mode is therefore reasonable for the investigated time-frame.

We thank the reviewer again for their efforts. The raised comments have been addressed in detail and we are confident that they resolve all previous concerns.

Reviewer 2:

In this paper, Schafer and co-authors use ab-initio QED simulation to investigate the vibrational strong coupling induced modifications of ground state chemical reactions. Through a direct quantum-electrodynamical density-functional simulation between a single molecule and a single mode of an optical cavity, they observed "multiple resonances behavior", which is, by matching the cavity frequency to the vibrational normal mode frequencies, the Si-C bond distance is reduced, thus, "prohibits" reaction from happening. Based on the simulations, the authors hypothesize that the cavity mode acts as mediator between different vibrational modes, which is the mechanism for the observed bond pretation mechanism. They further conclude that the vibrational energy localized in single bonds are critical for the reaction to be redistributed differently which ultimately inhibits the reaction. Despite its limitations (single molecule, no collective effect, single cavity mode, not enough statistics, and so on), to the best

of my knowledge, this is indeed the first work that treats the Ebbesen molecule through ab-initio calculations and using dynamics to investigate reactivities. In that sense, I do believe that the VSC community will benefit from such a work. I thus recommend this paper to be published on Nature communication, after the authors addressing the following comments.

We thank the reviewer for the very detailed and constructive comments. We appreciate and agree with the reviewers perspective that scientific research happens usually step-wise and the here presented work provides a significant step for the community.

(1) Connections with the previous work on vibrational energy redistribution. The mechanism proposed in this work, which is cavity mode acts as mediator between different vibrational modes, might be equivalent to the recent work in [J. Chem. Phys. 156, 014101 (2022)]. In that work, the cavity mode is coupled to a collection of vibrations, which are in turn coupled to the reaction coordinate, thus the cavity mode is effectively mediating energy transfer and influencing reaction rate constants. In that work, through both analytic theory and numerical simulations, it seems that when cavity frequency is tuned to match a frequency that depends on both vibrational frequency and the top of barrier frequency, the reaction rate constant is reduced. I believe that the mechanism is essentially the same as what presented here, which is the cavity mode mediating energy transfer among vibrational modes. In a related work [Angew. Chem. 2021, 133, 15661 -15668], this vibrational energy transfer mechanism is also explicitly explored and confirmed, with a much sharper resonance condition. A detailed comment on the connections between the current work and these two previous works will be extremely helpful.

We agree with the reviewer that it is certainly interesting to understand how this work compare with the existing literature. From our understanding, the conclusions drawn in [J. Chem. Phys. 156, 014101 (2022)] are closely related to [Nat. Commun. 12, 1315 (2021)] and show foremost that the concepts of the latter publication can be effectively transferred into the collective coupling regime under assumptions such as perfect alignment of the solvent. As discussed in the manuscript and the SI section 4, our QEDFT calculations reproduce the in [Nat. Commun. 12, 1315 (2021)] proposed effect at 86 cm^{-1} in Fig. 4 (b). However, the vibrational spectrum does not exhibit any relevant excitation in this domain, the effect seems therefore largely independent of the vibrational features of the system and can be traced back to the transition-state curvature. The existing experiments suggest that the resonant condition between an IR active excitation and the cavity mode control the chemical reactivity. This observation is more consistent with the remaining resonances in Fig 4 (b) which seem unconnected to the dynamical caging effect proposed in [Nat. Commun. 12, 1315 (2021)] and [J. Chem. Phys. 156, 014101 (2022)]. Why the here observed multitude of resonances, both at the transition-state curvature and some vibrational frequencies, has not been found before remains an open question. It seems likely that the model used by Huo et al. has a too small number of explicit degrees of freedom that go beyond the harmonic description in the Caldeira-Leggett approach. This would suggest that anharmonic components play a larger role than expected. Indications in this direction are supported by recent work [J. Phys. Chem. B 2021, 125, 8472], that shows

how VSC can alter the anharmonic interaction between vibrations. Alternatively, it would suggest that the exchange of vibrational energy is strongly memory dependent or demands additional modifications along the lines of Pollak-Grabert-Hänggi theory as proposed by Lindoy et al. [J. Phys. Chem. Lett. 2022, 13, 28, 6580-6586]. As can be seen in Fig 3, our calculations show a very mode-selective influence of the cavity that is partially coherent and partially incoherent. Let us point out that we describe only intra-molecular redistribution and that the precise redistribution depends greatly on the coupled vibrational mode (compare e.g. Fig 3 and SI-figure 5). **We repeat this aspect now in the conclusion and cite [J. Chem. Phys. 156, 014101 (2022)], [J. Phys. Chem. Lett. 2022, 13, 28, 6580-6586], and [J. Phys. Chem. B 2021, 125, 8472], in addition to Ref. [Nat. Commun. 12, 1315 (2021)].**

The question to which extend inter-molecular vibrational energy exchange could contribute to the mechanism of VSC mediated reactivity remains yet to be fully explored in our opinion. As pointed out by the reviewer (and discussed in the manuscript), Ref. [Angew. Chem. 2021, 133, 15661] showed that the cavity can facilitate the cooling of hot CO₂ molecules by providing an additional pathway for vibrational energy to relax from hot to cold molecules. Intuitively, this seems connected to our work but it remains to be investigated to which extend this observation can be transferred to the experimentally relevant reactions. We conclude, that especially the competition between intra- and inter-molecular redistribution processes should be the focus of future experimental work. A conclusive explanation of VSC mediated chemical reactivity will likely demand a collaborative effort between first-principles theory, model and analytical theory and experimental investigations.

(2) The limited numerical statistics. The main limitation of the current work is the number of trajectories, which is only 30, and only 8 of them are reactive and were used to analyze the results (Fig. 4b). Even though the time averaging gives a small enough error bar presented in Fig. 4b, the question remaining is how statistically meaningful for those 8 reactive trajectories, especially considering the very limited and specific phase space covered through their initial conditions. Can we trust the results presented in Fig. 4? By adding more trajectories, will all of these resonant features be washed away?

We acknowledge (also in the manuscript) that the statistical ensemble is small and we refrain from making quantitative predictions for this particular reason. It should be noted that the phase space that leads to the reaction is very small. We performed initial investigations using truly random initial velocities but noticed quickly that in the vast majority of trajectories the F⁻ ion is either repelled by the methyl groups or it attaches to the Si atom but does not induce the reaction. This is due to the fact that the system had to invest so much energy into re-organization that the reaction will stop at the intermediate step. The low reactivity is nicely embodied by the possible angle of attack for the F⁻. Our initial investigations found a second possible attack-angle at 60 degrees (compared to the Si-C axis) but the temperature necessary to see the reaction was more than 900 K, clearly not related to the experiment. The preferential sampling is now inspired by Bayesian-like approaches where a higher sampling density is assigned to areas that are known to be more relevant for the reaction. Specifically, we sampled 20 trajectories with a relative deviation of 20K around a reactive trajectory. While this certainly

does not exhaust the full reactive phase-space, it provides a good enough resolution to explore the basic mechanism of the reaction. This can be nicely seen from the Maxwell-Boltzmann distribution shown below. The histogram resembles the Maxwell-Boltzmann distribution at 300K and counts how many atoms possess a velocity that is within a given range.

Reactive trajectories are highlighted in red. A pronounced high-energy tail is necessary to undergo reaction, our dense sampling increases the probability for such a trajectory to be used within the calculation. It should be noted that even non-reactive trajectories can exhibit strong Si-C bond-stretchings which contributes additional information to our analysis in Fig. 4. It is interesting to look at the distribution when separated into groups of specific atoms or complexes, shown below.

Preferential sampling of the high-energy tail corresponds to a higher average temperature of the methyl groups while the attacking F^- is slower. This provides the methyl's with sufficient time and energy to re-arrange and facilitates building the pentavalent/intermediate state.

Adding more trajectories that are non-reactive would simply increase the weight of the reactant and thus lower the overall average bond distance. We do not expect any influence on the drawn conclusions.

We discuss this aspect now in more detail in the SI and provide the here shown Maxwell-Boltzmann distributions.

(3) *The VSC experiments reported rate constant suppression. The current paper reported a different quantity, which is the bond distance (Fig. 4). To what extent, these two quantities related? I understand intuitively, that makes sense, and the dip in Fig 4 that corresponds to the barrier frequency nicely corresponds to the rate constant reduction of the caging effect reported in Ref. 36. However, can the authors provide a more rigorous argument on the connections between them, which means a shortening of the equilibrium bond distance is guaranteed for a rate constant reduction.*

As pointed out by the reviewer, the connection between the chosen observables and experiment is intuitive but remains mathematically unclear. The high computational cost that limits the statistical ensemble size remains the main limitation that prevents us from simply applying the common techniques to estimate the reaction rate. Let us approach this interesting problem with the help of a 'Gedankenexperiment'.

For a defined reaction coordinate, let's assume for simplicity it would be represented by the Si-C bond distance, the average distance inside the cavity is smaller than outside the cavity for a given set of frequencies $R_{SiC}(\omega_{cav}) < R_{SiC}(no\ cav)$. The nuclear system is not static but features a temperature dependent average $R_{SiC}(T_{eff})$ as the anharmonic components result in thermal expansion. Reducing the average Si-C distance inside the cavity translates therefore in an effectively smaller temperature for the system, or in other words, the cavity acts as cooling device for the reactive coordinate. This is a simplified interpretation of our observations. If one would consider now classical transition-state theory, the reduction in effective temperature translates into an effective reduction of the rate $k_{cav} = Ae^{-E_A/k_B T_{eff,cav}}$ such that $k_{cav}(\omega_{cav})/k_{no\ cav} = Ae^{-E_A/k_B T_{eff,cav}}/k_{no\ cav} \approx e^{-E_A/k_B(1/T_{eff,cav}-1/T_{eff,no\ cav})}$ is strictly smaller than unity. In this highly simplified Gedankenexperiment, the cavity acts therefore as cavity-frequency dependent cooling device that reduces the rate and thus inhibits the reaction.

(4) *The theoretical approach that authors used is Ref. 40. Based on what I read, it seems to be an Ehrenfest dynamics that treats both nuclear and photonic DOF classically, and only treat the electronic DOF quantum mechanically. If so, this is then different than what is claimed in the main text "Our approach relies on the recently introduced quantum-electrodynamical density-functional theory (QEDFT) framework [23, 39-41] that enables the full description of electronic, nuclear and photonic degrees of freedom from first principles", because it sounds like the author treats everything at the DFT level of the theory, and the classical treatment is usually not considered as "first principle".*

The interpretation of the words 'ab initio' and 'first principles' are community specific and debatable. For us, both are synonymous and only mean that the calculations are 'unbiased', i.e., possess no free parameters (besides the light-matter coupling strength), and are immediately given by simply providing a molecular formula (e.g. C₆H₆ for benzene). In this specific case, the nuclei evolve according to the Ehrenfest forces from the electrons and photons. The

photonic system evolves according to Maxwell's equations, which is the lowest-order approximation in QEDFT. While the interaction is classical, the approach is still capable to provide classical correlation between all DOF. We transparently discuss the used approximation in Section 1, see e.g. "The correlated evolution of electronic, nuclear, and photonic system is described by quantum-electrodynamical density-functional theory using Ehrenfest's equation of motion for nuclear and Maxwell's equation for photonic degrees of freedom [40] (details in Materials and Methods IV)."

(5) What is the difference between this approach in Ref. 40 with the "adiabatic dynamics", which propagates the classical nuclear and photonic trajectories on the adiabatic electronic ground surface, such as used in those earlier studies in Ref. 36, in [J. Chem. Phys. 156, 014101 (2022)] and in [Angew. Chem. 2021, 133, 15661 - 15668]. Further, the authors need to provide more details of this method in either the method section or SI to make the current paper self-contained.

In this manuscript we are using a time-dependent version of the quantum-electrodynamical density functional theory to perform the actual calculations as has been published in Ref. 40. In practice we are treating the electron-nuclear dynamics on the level of the Ehrenfest dynamics and the matter-photon dynamics on the level of Maxwell's equations. While the Ehrenfest approach has been demonstrated to include some effects of nonadiabaticity (see e.g. J. Chem. Theory Comput. 2009, 5, 4, 728-742 for a discussion on this topic), the mentioned work (Ref.36, J. Chem. Phys. 156, 014101 (2022), and Angew. Chem. 2021, 133, 15661 - 15668) are based on molecular dynamics, thus adiabatic dynamics. In addition, our approach computes the electronic structure from first principles, while Refs. 36 and J. Chem. Phys. 156, 014101 (2022), use a model system and Angew. Chem. 2021, 133, 15661 - 15668 a classical force field to describe the electronic structure. We find that the first-principle description is necessary to find the correct pathway of the chemical reaction.

To clarify this discussion, we have added a more detailed description of the computational approach and added all mentioned references.

(6) Besides the single molecule limitation, the other major limitation of the current work is that it does not consider cavity loss. Cavity loss explicitly presents in the VSC experiments with microcavities, and it has been theoretically shown to further facilitate the "energy transfer mechanism" between vibrational modes and the cavity mode, in [J. Chem. Phys. 156, 014101 (2022)]. A comment on this will be very helpful.

To the best of our knowledge, loss has been identified as relevant for photo-chemical reaction and especially for plasmonic strong coupling where a real external excitation is sent into the set-up. Plasmonic systems feature extremely short lifetimes, often on the order of a few fs, due to the quick Landau-damping and hot-carrier generation. This is conceptually very different to the here discussed vibrational strong-coupling in the dark (no illumination) for a ground-state chemical reaction. Furthermore, the experimentally used infrared-cavities feature much

higher quality-factors than plasmonic systems and can be considered as coherent on time-scales less than a ps (as investigated in this work). To be specific, Thomas et al. [28] found a FWHM of the cavity eigenmode of 30 cm^{-1} which corresponds via $\tau_{cav} = 1/\Delta f = 1/(\tilde{\nu}c) = 1/(15 \text{ cm}^{-1} \cdot 2.998 \cdot 10^{10} \text{ cm/s})$ to a lifetime of 2.22 ps, which is about 4 times the reaction-time. Cavity-loss is thus much less likely to play a significant role. Nevertheless, in order to support our intuition, **we added losses to the cavity mode in our simulations, mention its effect in the main text, discuss its influence in the SI, and illustrate its relevance below.**

We introduce a friction of the form $2\gamma\dot{q}_\alpha(t)$ to the EOM of the photonic mode and recalculate all reactive trajectories for $\omega_c = 571 \text{ cm}^{-1}$. The used friction values are multiples $\gamma = \{0, 1, 2, 4, 8\} \cdot \gamma_{exp}$ of the experimentally observed value $2\gamma_{exp} = 30 \text{ cm}^{-1}$ or 0.000137 in atomic units.

The influence of photonic friction on the average Si-C bond length depends on time. For short times, we see an increase of the inhibiting effect of the cavity. This short-time effect tends to decrease over time and leads to only minute changes in the time-averaged Si-C distance. Our approximation of a loss-less cavity mode is therefore reasonable for the investigated time-frame.

(7) I would recommend the authors to carefully summarize what agrees with the experimental results and what disagrees with them. For example, the theoretical prediction has a much

broader resonance compared to the sharper resonance observed in the experiments, and the experimental frequency at 856 cm⁻¹, which also has a very strong resonance suppression of the rate is missing in the current theoretical prediction.

While we discuss those aspects throughout the manuscript, we agree with the referee that the reader would benefit from such a summary in the conclusion. **We reformulated the conclusion and summarize now briefly all agreeing and disagreeing features.**

We thank the reviewer again for the constructive comments and hope that the response is able to account for all questions.

Reviewer 3:

The article by Schafer et al. reports QEDDFT/Ehrenfest simulations of an organic silicon deprotection reaction under the influence of a boson mode representing an optical microcavity. This reaction was first studied experimentally in a Fabry-Perot cavity under conditions of collective strong coupling by Thomas et al. The simulation results are valuable contributions to the field. However, as explained in detail below, some of the main claims made in the article are highly questionable when viewed in light of recent works in this field.

We thank the reviewer for their constructive response and the overall assessment that our work represents a valuable contribution to the field. Some of the comments below can be well accounted for with a few more details that we added to the manuscript.

1. Previous articles have already shown that the cavity can act to suppress a reaction when the light-matter coupling is many orders of magnitude greater compared to experiments in Fabry-Perot cavities [see e.g., <https://journals.aps.org/prx/abstract/10.1103/PhysRevX.9.021057>, <https://pubs.acs.org/doi/pdf/10.1021/acs.jpcllett.2c00974>, <https://www.sciencedirect.com/science/article/abs/pii/S0301010419311140>, <https://aip.scitation.org/doi/10.1063/5.0006472>]. The challenge, as laid out in e.g., the PRX of Feist et al (as well as in several other works) is to explain why the cavity effect persists under conditions of collective strong coupling, where a large number of molecules couple to the field, the polaritons constitute a tiny fraction of the total number of states, and the single-molecule light-matter coupling is extremely weak (orders of magnitude weaker than those used by the authors). In a long complicated paragraph on page 4, the authors present a brief discussion of the above point, but it remains very unclear to this reviewer why the effect they reported is expected to persist when a realistic value of the single-molecule light-matter coupling is several orders of magnitude smaller than the valued used in the simulations.

To the best of our knowledge, no previous publication has been able to recover the experimental observations for vibrational strong coupling. Especially the resonant condition remained elusive. The very recent addition by Sun and Vendrell [JPCL 2022 13 (20), 4441], which has been published one year after our initial submission, is in our opinion the only publication, besides our work, that is able to capture similar effects even if their chemical reaction is much simpler and not related to any experiment available in the literature. Our work is thus not only the first *ab initio* study that reports the resonant effect, it also represents the first work that

describes explicitly the experimentally investigated reaction, contributing a vital study for the future development of polaritonic chemistry.

Nevertheless, we agree with the reviewer that the question "How can it be that the reaction for single molecules is affected when the alleged contribution per molecule is vanishingly small?" is an important aspect of polaritonic chemistry. The here chosen coupling strength is larger than we expect from the experiment and we show in the SI that the effect is decreasing when reducing the coupling strength. Three aspects should be considered in this context:

(1) We would like to stress that the widely believed assumption that the single molecule is only contributing with a factor $1/N$ is not at all evident during a chemical reaction. This picture is only reliable for simplified/static systems which consists of identical molecules. As pointed out in references [5, 45, 46, <https://doi.org/10.1021/acs.jpcclett.2c01169>], this does not imply that a molecule that undergoes a chemical reaction cannot be substantially affected by the ensemble. In [<https://doi.org/10.1021/acs.jpcclett.2c01169>], it is shown that proton-tunneling can depend non-trivially on the number of emitters. Depending on the fundamental coupling strength and the number of emitters, the tunneling can be facilitated or inhibited. In any case, the intensity of those effects can even rise with increasing N and will become negligible only for extremely large N . The assumption that a reacting molecule is only feeling a tiny fraction of the collective effect is therefore too simplistic. Furthermore, the precise number of collectively coupling molecules in experiment is unknown and has been only estimated via simple theoretical models.

(2) We acknowledge in the manuscript, for instance in the sentence "While intermolecular interactions could contribute to the energy redistribution responsible for the cavity mediated effect on chemical reactivity, the agreement between our theoretical predictions and experimental observations suggests that the mechanism will remain qualitatively similar.", that intermolecular interaction channels remain possible candidates. Importantly, even if this cannot be excluded and a fully conclusive understanding of vibrational strong coupling mediated chemistry is therefore still absent, our results are qualitatively so consistent with experiments that it seems highly unlikely that the basic mechanism is not at least related. We would like to point out that we clearly discuss this aspect throughout the manuscript and even conclude the manuscript with "Handles such as symmetry and coherence could elucidate to which extend inter- and intramolecular energy redistribution contribute to vibrational strong-coupling. How such kinetically driven effects can persist into the realm of realistic ambient conditions for large ensembles of reacting molecules thus remains a key question necessitating further theoretical and experimental investigations where the present findings provide the seed to guide them."

(3) The currently available QEDFT implementation in the Octopus code is limited for a number of reasons. Each trajectory, for a single molecule, demands roughly 18000 core-hours, a

non-negligible cost that prevents us from using hundreds of thousands of trajectories. This implies that the cavity has to noticeably affect the few trajectories in the short time of the reaction, i.e., we need a strong effect over short times and thus a strong light-matter coupling. The used QEDFT approach contributes however also clear strengths that complement existing investigations. Those are foremost the true *ab initio* treatment and the fully correlated motion of all constituents.

We acknowledge at many places in our manuscript that the single-molecule coupling approach is a limitation that has to be carefully lifted in the future. However, our manuscript provides still a significant development step in the endeavour to understand polaritonic chemistry – as becomes clear from the fact that we recover for the first time critical experimental features using an *ab initio* approach. That this development step does not provide the ultimate conclusion to such a complex problem seems intuitive and we acknowledge the need for further theoretical and experimental studies.

The paragraph mentioned by the reviewer has been reformulated and simplified. We are confident that the new version is capable to transfer this perspective in a more concise way.

2. In the experiments where reactivity was changed by the cavity, 'off-resonance' only refers to the cavity mode with zero in-plane wave-vector magnitude ($q = 0$), but in general this means that some other cavity mode with q different from zero is on-resonance with the targeted molecular vibration. Given that these are thermal reactions, there is no obvious reason why a cavity resonance at $q = 0$ with some molecular vibration would be more efficient at suppressing a chemical reaction than a cavity mode with q different from 0. In the authors' simulations, as in other recent papers, a "resonant effect" is observed, but that could be a mere byproduct of the very extreme assumption that all cavity effects can be described with a single-photon mode at $q = 0$. Whether such resonance effect would persist under a more realistic treatment of the optical cavity that accounted for cavity-matter resonance at $q = 0$ is a question that the authors have completely ignored. In fact, there is no discussion of their central assumption that the cavity can be described with a single photon mode when the existence of a continuous spectrum of q different from zero modes in a planar Fabry-Perot cavity is well known. Furthermore, polaritons at incidence angles different from zero are routinely observed and play a key role in photoluminescence, Bose-Einstein condensation, thermal emission, etc.

We thank the reviewer for this important comment. It is indeed counter-intuitive that only $q = 0$ modes should be able to influence chemical reactivity, many propagating/guided waves are allowed in a material that might just as well affect a chemical reaction. Our simulations are limited to the $q = 0$ case and we acknowledge that a more detailed discussion of this aspect was missing. The additional discussion provides in our opinion strong arguments for why our approach is adequate.

Most materials support propagating/guided modes. This feature is sometimes referred to as 'bulk polaritons', i.e., the material possesses 'intrinsic polaritons'. The cavity is now in-

roducing a lower bound for the allowed frequencies as the boundary-conditions ensure that $k_z = n\pi/L_z$ (for a cavity with parallel mirrors in the x-y plane). Modes with a large in-plane momentum are qualitatively similar to those in bulk, only the modes with q near 0 deviate significantly inside a Fabry-Perot cavity. Vurgaftman et al. [J. Chem. Phys. 156, 034110 (2022); <https://doi.org/10.1063/5.0078148>] showed with the help of classical density-of-state arguments that the effects of a planar cavity, compared to bulk polaritons (or those in thin slabs), are negligible when q deviates significantly from 0. Furthermore, the influence of the cavity decreases with increasing length, as we expect intuitively. This implies that chemical reactivity *via the cavity* will only appear for approximately $q = 0$ and low harmonics n . While one will surely still observe polaritons for $q > 0$, their relevance compared to the bulk polaritons is quickly diminishing with increasing q . This is rather intuitive, the higher q the more light will be simply propagating along the sample. A molecule that undergoes a reaction will have the opportunity to coherently interact with the cavity excitation for $q = 0$ while it will feel dominantly guided/propagating modes that exist in almost the exact same form without the cavity for $g > 0$. A real difference to the situation without the mirrors is thus only expected for $q = 0$.

We discuss this aspect now in the manuscript.

Given the above arguments, while I agree with the authors that the 'resonant effect' is 'theoretically elusive but experimentally critical', I respectfully disagree that their simulations have explained the mechanism underlying this effect. In summary, while this is an interesting manuscript, I don't think it provides resolution to the two main challenges of polariton chemistry described above. Therefore, I suggest publication in a more specialized journal (after reformulation of some of the conclusions).

We thank the reviewer for their constructive comments and hope that the response provides a clearer picture of our perspective. To conclude, we strongly believe that the present discussion is tainted by assumptions that are drawn from simplified models which are no longer adequate in this regime. The manuscript acknowledges and discusses limitations of this **first** *ab initio* work and yet, we clearly recover important experimental features. In our opinion, this suggests that the fundamental mechanisms for vibrational strong coupling is closely related to the here observed effect. It should be emphasized that our work is the only available publication that simulates the same experimentally studied chemical reaction from first principles and finds resonant modification under vibrational strong coupling. Therefore this manuscript provides an important development step for polaritonic chemistry.

Yours sincerely,

Christian Schäfer,
on behalf of all authors.

REVIEWERS' COMMENTS

Reviewer #2 (Remarks to the Author):

I appreciate the efforts from the author to address my concerns and I think the revised version is suitable for publication in Nat. Comm.

Reviewer #3 (Remarks to the Author):

1. On the issue that their results are substantially dependent on the many-order-of-magnitude larger theoretical coupling strength relative to experiments, the authors agree that they can't answer the key question of why the reactivity would be dramatically changed when the (real) single-molecule light-matter interaction strength is so small in microcavities. Important context has been added, which is appreciated.

2. None of my criticisms are related to "simple/static models" and much less to whatever "1/N" issue is raised by the authors. I agree that the question is very open and it is unclear how collective effects change the simplified single-molecule system studied here (although <https://arxiv.org/abs/2206.08937> suggests a pessimistic view of what will happen when one considers a realistic disordered ensemble). My point is simply that the order of magnitude of the coupling strength is too large to make meaningful statements about experiments conducted in the thermodynamic limit. Therefore the main puzzles in the field remain unresolved. I don't expect the authors to provide the answers to all questions involved in these complex experiments, and do understand the computational limitations, which should therefore limit the scope of the stated conclusions.

3. Work by Vurgaftman et al. that suggests the molecular density of states is mostly affected by a cavity when the resonance with the molecular ensemble is at $q = 0$ is cited to justify the assumption that a single boson mode can accurately describe a microcavity. Aside from the fact that the calculations of Vurgaftman et al. were done within a purely classical framework (what I suppose the authors would classify as a "simplified/static model" if it were not making a point that they support), when here the authors highlight their "ab initio fully quantum" model, a more urgent aspect to consider is that the observed effects on density of states reported by Vurgaftman were almost negligible (with normalized difference in the tail of molecular density of states with order of magnitude 10^{-9}), so it is hard to take it as a rigorous argument that such negligible changes justify working with exclusively with the $q = 0$ mode. This is especially true in light of recent *quantum-mechanical* work such as <https://journals.aps.org/prx/abstract/10.1103/PhysRevX.10.041027>, which shows in incontrovertible terms that a realistic model of the microcavity (and strong coupling happening at different values of the incidence angle) is a key requirement for describing polaritonic effects on large thermal ensembles of emitters. In summary, it is far from obvious that the argument by Vurgaftman et al. applies with the generality implied by the authors (it does not), nor that it is particularly relevant in the context of polariton effects on reactivity since correlation does not equal causation and the differences in $q = 0$ effects on the molecular density of states vs. q different from zero are essentially negligible (10^{-9} effect happening on the tail of the normalized molecular density of states).

With this said, I think the manuscript reads much better than in the prior version, and believe the authors have made a compelling case that it merits publication in Nature Communications, so I would have no problem recommending publication if (a) the issue in item 3 is also addressed in light of the given reference and the points made there, and (b) the statements in the abstract are attenuated to reflect the fact that the main unresolved questions in this field remain open (the impression given by the abstract is very different from what has been acknowledged by the authors).

Reviewer 2:

I appreciate the efforts from the author to address my concerns and I think the revised version is suitable for publication in Nat. Comm.

We thank the reviewer for their valuable and constructive comments during the review process and are grateful for the final approval.

Reviewer 3:

1. On the issue that their results are substantially dependent on the many-order-of-magnitude larger theoretical coupling strength relative to experiments, the authors agree that they can't answer the key question of why the reactivity would be dramatically changed when the (real) single-molecule light-matter interaction strength is so small in microcavities. Important context has been added, which is appreciated.

We thank the reviewer for their constructive comments and the overall fruitful discussion. As the reviewer correctly states, we discuss in fair terms the strength and limitations of our theoretical description of the chemical reaction under strong coupling conditions. What we can say with confidence is that we observe a change of the chemical reactivity under strong coupling that is qualitatively consistent with the experiment. In order to resolve this for a single molecule, we require an amplified effective coupling where relative changes of it show a similar trend as in experiment.

2. None of my criticisms are related to "simple/static models" and much less to whatever "1/N" issue is raised by the authors. I agree that the question is very open and it is unclear how collective effects change the simplified single-molecule system studied here (although <https://arxiv.org/abs/2206.08937> suggests a pessimistic view of what will happen when one considers a realistic disordered ensemble). My point is simply that the order of magnitude of the coupling strength is too large to make meaningful statements about experiments conducted in the thermodynamic limit. Therefore the main puzzles in the field remain unresolved. I don't expect the authors to provide the answers to all questions involved in these complex experiments, and do understand the computational limitations, which should therefore limit the scope of the stated conclusions.

We understand the reviewers perspective but would like to briefly clarify why we are confident that our work plays an integral role in the theoretical understanding of polaritonic chemistry. From our perspective, in order to draw any relevant conclusions related to the experiments the realistic chemical reaction is the important starting point. The extremely complex dynamic of the experimental system, for which many aspects such as the precise number of collectively coupling molecules are unknown, requires a methodical approach to resolve it. We decided here to keep the most important aspect, i.e., the chemical reaction, intact while using a more minimalist but working description for the environment. We openly discuss shortcomings and remaining questions, especially how intermolecular effects could modify the observations, but also illustrate a potential mechanism that is reasonable and qualitatively consistent with the experiment. We entirely agree with the reviewer that further work will be required for

a holistic understanding and we are confident that our work will substantially contribute to it. Lastly, the theoretical efforts should be accompanied with additional experiments and we conclude with possible suggestions that could further illuminate polaritonic chemistry.

3. Work by Vurgaftman et al. that suggests the molecular density of states is mostly affected by a cavity when the resonance with the molecular ensemble is at $q = 0$ is cited to justify the assumption that a single boson mode can accurately describe a microcavity. Aside from the fact that the calculations of Vurgaftman et al. were done within a purely classical framework (what I suppose the authors would classify as a "simplified/static model" if it were not making a point that they support), when here the authors highlight their "ab initio fully quantum" model, a more urgent aspect to consider is that the observed effects on density of states reported by Vurgaftman were almost negligible (with normalized difference in the tail of molecular density of states with order of magnitude 10^{-9}), so it is hard to take it as a rigorous argument that such negligible changes justify working with exclusively with the $q = 0$ mode. This is especially true in light of recent *quantum-mechanical* work such as <https://journals.aps.org/prx/abstract/10.1103/PhysRevX.10.041027>, which shows in incontrovertible terms that a realistic model of the microcavity (and strong coupling happening at different values of the incidence angle) is a key requirement for describing polaritonic effects on large thermal ensembles of emitters. In summary, it is far from obvious that the argument by Vurgaftman et al. applies with the generality implied by the authors (it does not), nor that it is particularly relevant in the context of polariton effects on reactivity since correlation does not equal causation and the differences in $q = 0$ effects on the molecular density of states vs. q different from zero are essentially negligible (10^{-9} effect happening on the tail of the normalized molecular density of states).

We do not question the value of simple classical descriptions that facilitate the understanding of a specific aspect, merely the reliability of oversimplified models for chemical reactivity. The reviewer is entirely correct that the effect observed by Vurgaftman et al. is small, **we added such a statement to the manuscript**. It should be noted that changes often exponentially influence a chemical reaction and the size of the described effect remains secondary as the classical model lacks many important details. Irrespective therefore, the study aimed to answer the question why $q = 0$ should be a meaningful condition and, in our opinion, succeeded to provide a reasonable explanation. The work mentioned by the reviewer describes extended systems on the verge to a macroscopic ferro-electric phase, i.e., the cavity couples to macroscopic phononic modes with sizeable in-plane momentum. We see at best a very remote relation between the systems as the molecular vibrations are entirely localized in the absence of the cavity, which is in stark contrast to a macroscopic slab of a solid. Furthermore, the mentioned work is purely theoretical and we are not aware of any experimental confirmation. The $q = 0$ limit has been experimentally validated by multiple authors.

With this said, I think the manuscript reads much better than in the prior version, and believe the authors have made a compelling case that it merits publication in Nature Communications, so I would have no problem recommending publication if (a) the issue in item 3 is also

addressed in light of the given reference and the points made there, and (b) the statements in the abstract are attenuated to reflect the fact that the main unresolved questions in this field remain open (the impression given by the abstract is very different from what has been acknowledged by the authors).

We thank the reviewer for their constructive comments and the fruitful debate. **We adjusted the abstract to better reflect the single mode approximation and added the in item 3 suggested comment regarding the size of the effect observed by Vurgaftman et al.**

Yours sincerely,

Christian Schäfer,
on behalf of all authors.